# Neuronal selectivity for stimulus information determines prefrontal LFP gamma power regardless of task execution

Balbir Singh [1 ✉], Zhengyang Wang[2] & Christos Constantinidis[1,2,3]

Local field potential (LFP) power in the gamma frequency is modulated by cognitive variables during task execution. We sought to examine whether such modulations only emerge when task rules are established. We therefore analyzed neuronal firing and LFPs in different prefrontal subdivisions before and after the same monkeys were trained to perform cognitive tasks. Prior to task rule learning, sites containing neurons selective for stimuli already exhibited increased gamma power during and after the passive viewing of stimuli compared to the baseline period. Unexpectedly, when the same monkeys learned to maintain these stimuli in working memory, the elevation of gamma power above the baseline was diminished, despite an overall increase in firing rate. Learning and executing the task further decoupled LFP power from single neuron firing. Gamma power decreased at the time when subjects needed to make a judgment about whether two stimuli were the same or not, and differential gamma power was observed for matching and nonmatching stimuli. Our results indicate that prefrontal gamma power emerges spontaneously, not necessarily tied to a cognitive task being executed.

[1] Department of Biomedical Engineering, Vanderbilt University, Nashville, TN 37235, USA. [2] Neuroscience Program, Vanderbilt University, Nashville, TN 37235, USA. [3] Department of Ophthalmology and Visual Sciences, Vanderbilt University Medical Center, Nashville, TN 37232, USA. ✉email: balbir.singh@vanderbilt.edu

Cognitive processes modulate simultaneously the activity of a large number of neurons and produce synchronized neural inputs and outputs, reflected in the rhythmicity of extracellular field recordings, including EEG, MEG, and LFP[1,2]. Rhythmicity in the gamma frequency band is thought to be generated by correlated bottom-up inputs that synchronize a population of pyramidal neurons followed by a volley of inhibition by interneurons, which thus produces rhythmicity in the field potentials[3]. In the prefrontal cortex, LFP gamma band power increases in the delay periods of working memory tasks and is tuned for information held in memory[4–8] and more so in electrode sites where neurons encode information about the stimuli being represented in memory[9,10]. Theories of working memory have therefore proposed that gamma band LFP rhythmicity is reflective of the underlying neural activity that allows maintenance of information, and provides better fidelity than neuronal firing rate itself[10,11].

An implicit assumption of these theories is that gamma band rhythmicity emerges as a result of active maintenance of working memory. However, recent studies suggest that robust gamma-band power, in the 45–100 Hz range, during and after a stimulus presentation compared with the baseline, may be generated absent execution of a cognitive task, in subjects naïve to training, while subsequent training in fact decreases its magnitude[12,13]. This result is puzzling considering that activity of individual neurons is known to be affected by learning to perform such tasks in that a greater number of neurons are activated by stimuli after training[14,15], their firing rate generally increases[16], and so does their selectivity for stimuli[17]. All of these effects would seemingly predict enhanced gamma oscillations[18,19] after training and execution of a task, according to the theories linking informative neurons with gamma band power during working memory[10,11,20].

To resolve the question, it is essential to understand jointly the properties of neurons and LFPs and compare them in subjects performing working memory tasks or not. We thus analyzed firing rates and LFP gamma power from the same prefrontal subdivisions before and after monkeys were trained to perform spatial and shape working memory tasks. Such training is well known to alter neuronal responses, an effect evident across a range of methodologies[21]. In doing so we sought to reveal the constituent neural changes related to the generation and modulation of gamma-band power in LFPs before and after training to perform working memory tasks and test whether increased gamma power is the result of task execution.

## Results

We recorded LFP activity from the lateral prefrontal cortex of three monkeys trained to perform a match–nonmatch working memory task and from the same monkeys prior to training, while they viewed stimuli passively (Fig. 1a, b). Two stimulus sets were used in these experiments, one varying the spatial location of a white square (spatial set), and one involving different geometric shapes (shape set). The monkeys had to observe two stimuli presented in sequence and, after a delay period, determine whether they the appeared at the same location and were the same shape or not. Neuronal and LFP data were obtained from the anterior-dorsal, posterior-dorsal, mid-dorsal, anterior-ventral, and posterior-ventral subdivisions of the prefrontal cortex (Fig. 1c). A total of 1074 neurons from 380 electrodes were recorded during passive viewing of the spatial stimulus set, prior to training, and 1179 neurons from 500 electrodes were recorded after training and performance of the spatial working memory task. An additional 639 neurons from 224 electrodes were recorded during passive viewing of the shape stimulus set, prior

to training, and 867 neurons from 351 electrodes were recorded after training and performance of the shape working memory task. Areal breakdown of neurons and electrodes is shown in Supplementary Table 1, with posterior-dorsal, mid-dorsal, and posterior-ventral areas best represented. Each electrode yielded one LFP signal; 2–3 neurons were typically isolated from each electrode.

We classified sites as selective when all single neurons recorded from the site exhibited significant selectivity for a stimulus set (evaluated with a 1-way ANOVA, at the $\alpha = 0.05$ significance level). Sites were classified as nonselective when none of the single neurons recorded exhibited selectivity. The remainder of the sites, which involved recordings from some selective and some non-selective neurons were deemed as partially-selective (Fig. 1d). Although sites were classified based on selectivity alone, firing rate also differed systematically between neurons in these sites, with selective sites generally being more responsive to stimuli (Fig. 2a).

**γ-power is enhanced in selective sites regardless of task**. Previous reports have suggested that gamma power is enhanced in sites selective for the stimulus information being used for a working memory task[10]. We sought to test whether this relationship held equally true both before and after the stimulus information is task relevant. We therefore grouped the power spectrograms from the selective, partially-selective, and non-selective sites, when the subjects were tested with the spatial stimulus set, after training (Fig. 2d). We focused exclusively on gamma power, defined a priori as power in the range of 45–100 Hz, as this was the critical frequency band identified in these earlier studies. We sought to test how robust power changes around task events were across sites, so we calculated power in five discrete task epochs (fixation period, cue presentation, first delay, sample presentation, second delay), after subtracting the power in the inter-trial interval, at each frequency. We then averaged power computed in each interval for all trials obtained from each electrode and treated it as a single observation. A different baseline between animals unequally represented between conditions may artifactually appear as an effect of the treatment[22]. To avoid such a bias, we adopted a mixed-effects model, with fixed factors the selectivity status of each site (selective, nonselective, or partially-selective) and the task epoch and added a random effect term for different animals. Indeed, we found that gamma power was highest in the selective sites after training ($F_{2,2485} = 23.13$ for main effect of site selectivity, $p < 0.001$). Gamma power was significantly higher in selective sites compared with the other two groups, and higher in the partially-selective compared with the nonselective sites (post-hoc test, evaluated at the $\alpha = 0.05$ significance level). Robust gamma power was evident in selective sites when neuronal selectivity was defined for different stimulus presentation and delay epochs (Supplementary Fig. 1). The full distribution of gamma power values is shown Supplementary Fig. 2. Additionally, gamma power was the highest during the fixation and cue epoch of the task; a significant main effect of task epoch was present in the model ($F_{4,2483} = 8.2$, $p < 0.001$). The time course of gamma power modulation was quite distinct from the envelope of firing rate activity (Fig. 2a–c).

This result confirmed higher gamma power for selective sites, in agreement with prior studies. Importantly, when we repeated the same analysis for the pretraining data (Fig. 2e), we saw the same relationship: gamma power was highest for selective sites, even when the identity of the stimulus was irrelevant for the task, in naïve subjects only trained to fixate. A significant main effect of site selectivity status was again present ($F_{2,1883} = 20.6$, $p < 0.001$).

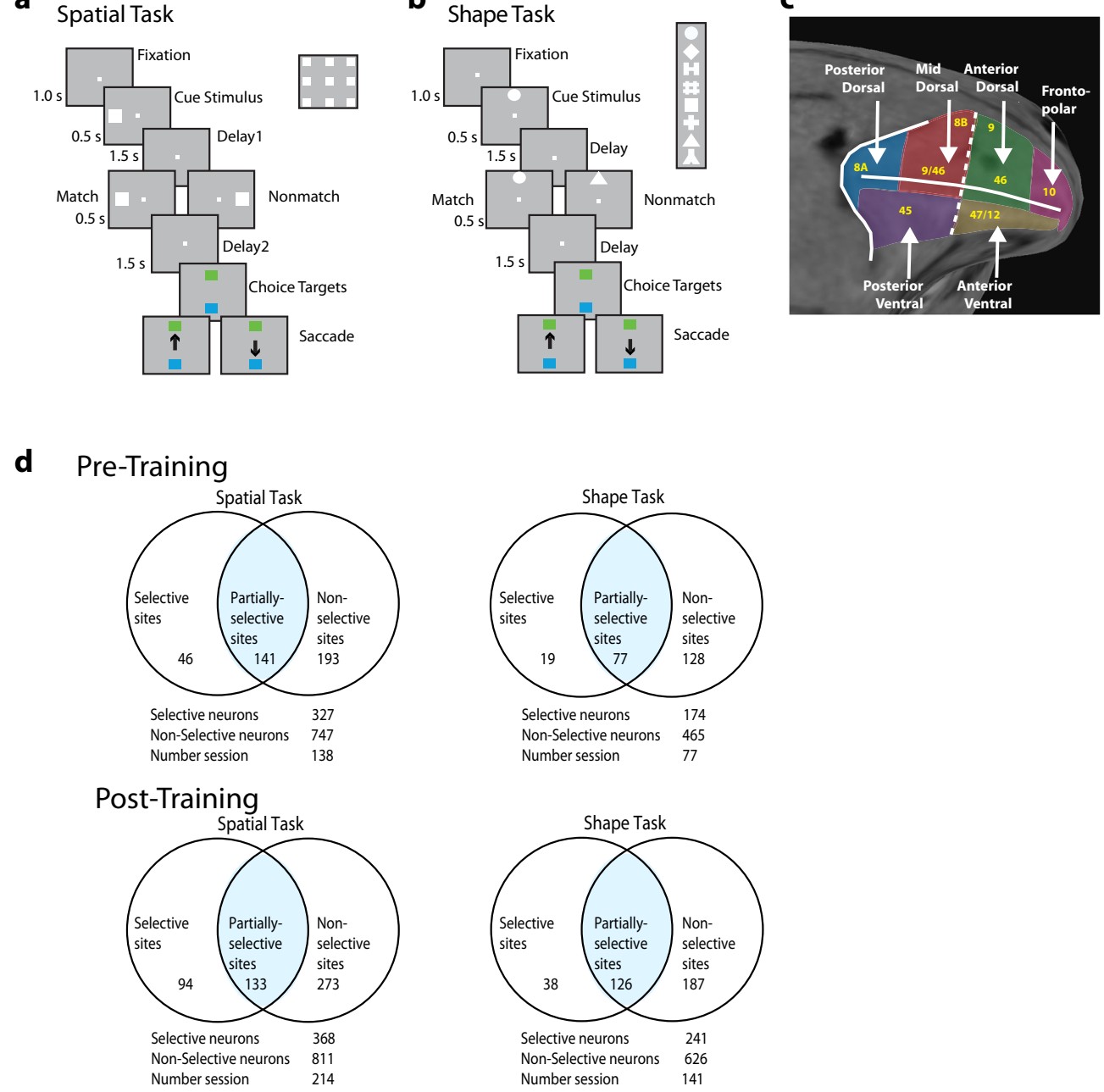

**Fig. 1 Task and recording sites. a** Sequence of frames indicates events in the spatial match-to-sample task. The animals were required to maintain center fixation throughout both active and passive task trials. At the end of active tasks trials, however, monkeys were required to make a saccade to a green target if the stimuli matched or to a blue target if the stimuli did not match. **b** Shape feature match-to-sample task, 8 possible shapes in a session shown in the inset. **c** Anatomical location of areas where recordings were made in the lateral prefrontal cortex. **d** Percentage of electrode sites for which all isolated neurons had significant selectivity for the location or shape of the stimulus (selective sites); for which some neurons had selectivity (partially selective); and for which no neurons had selectivity for stimuli (non-selective).

However, the predominant effect of training was a decrease in gamma power. To compare changes before and after training, we used a 3-way model, with fixed-effect terms comprising the pre- or post-training condition; site selectivity; and task epoch; and including again individual monkeys as a random effect. A highly significant decrease in gamma power was evident after training ($F_{1,4369} = 86.3$, $p < 0.001$). Data from the three monkeys separately are shown in Supplementary Fig. 3.

We further examined how gamma power differed between different prefrontal subdivisions and saw that gamma power was most evident in the subdivisions that exhibited the most

selectivity for the spatial stimulus set, namely the mid-dorsal and to a lesser extent the posterior dorsal prefrontal cortex (Fig. 3a, b). Firing rate of neurons in these areas is shown in Supplementary Fig. 4. To test how robust power changes around task events were across sites before and after training, we compared power values across conditions using a 3-way model, with fixed-effects that included the pre- or post-training condition; site selectivity; and task epoch. This analysis was performed separately for power in each of each region (i.e. a separate 3-way model was computed). A significant main effective of selectivity status was present in the mid-dorsal prefrontal

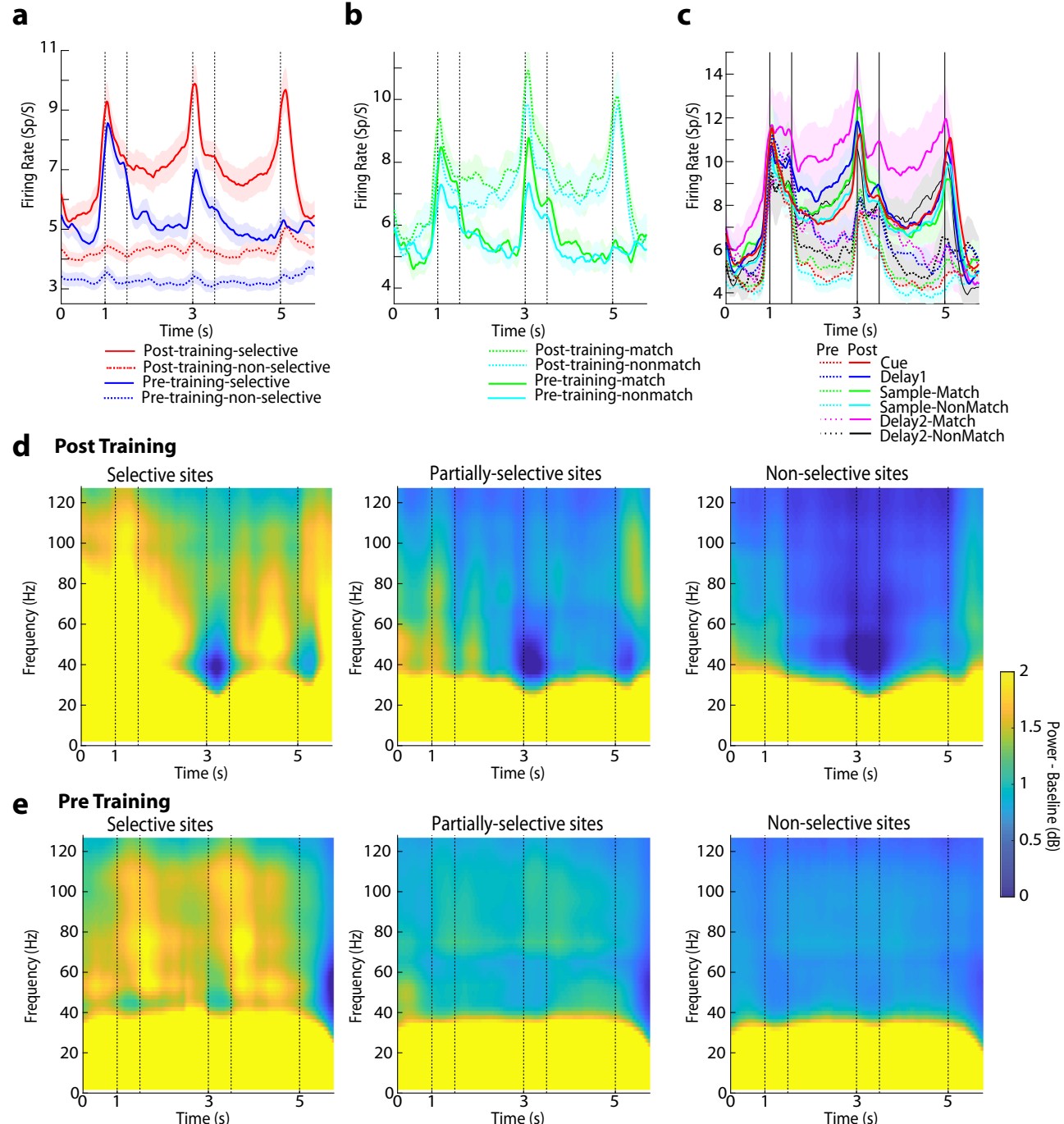

**Fig. 2 Firing rate and spectral power in the spatial task. a** Mean firing rate is averaged across all stimuli for sites classified as selective and non-selective, for different prefrontal subdivisions. Averaged neuronal spiking is shown separately for selective (N = 327 and 368 prior to training and after training, respectively) and non-selective sites (N = 747 and 811, respectively). Horizontal lines delineate the five task epochs starting with fixation (0-1 s), then two stimulus presentations (at 1-1.5 and 3-3.5 s), followed by corresponding delay periods (at 1.5-3 and 3.5-5 s) and choice target presentation (at 5 s).
**b** Averaged neuronal spiking shown separately for match and nonmatch trials from selective sites (N = 327 and 368 prior to training and after the training, respectively) **c** Averaged neuronal spiking is shown separately for neurons selective at different task epochs. Neurons selective for the first stimulus (N = 183); for the match and non-match (second) stimulus, (N = 129, 144 respectively) prior to training; for the first stimulus (N = 209) and for the second stimulus (N = 140, 169, respectively) after training. Neurons selective during the first delay period (N = 83) and for match and non-match during the second delay periods (N = 67, 54, respectively) prior to training; for first delay (N = 138) and for match and non-match during the second delay periods (N = 83, 99) after training. **d** Mean LFP induced spectral power recorded with the spatial stimulus set from the prefrontal cortex after and before training. Spectral power from the selective sites (left column), partially-selective sites (middle column) and non-selective sites (right column) is shown. Power is plotted as a function of time, after subtracting the mean power computed in the inter-trial interval at each frequency. Results after training (*n* = 13654 trials for selective sites, 16750 trials for partially-selective sites and 34402 trials for non-selective sites). **e** Results prior to training (*n* = 6886 trials, 22206 trials, and 27949 trials, respectively).

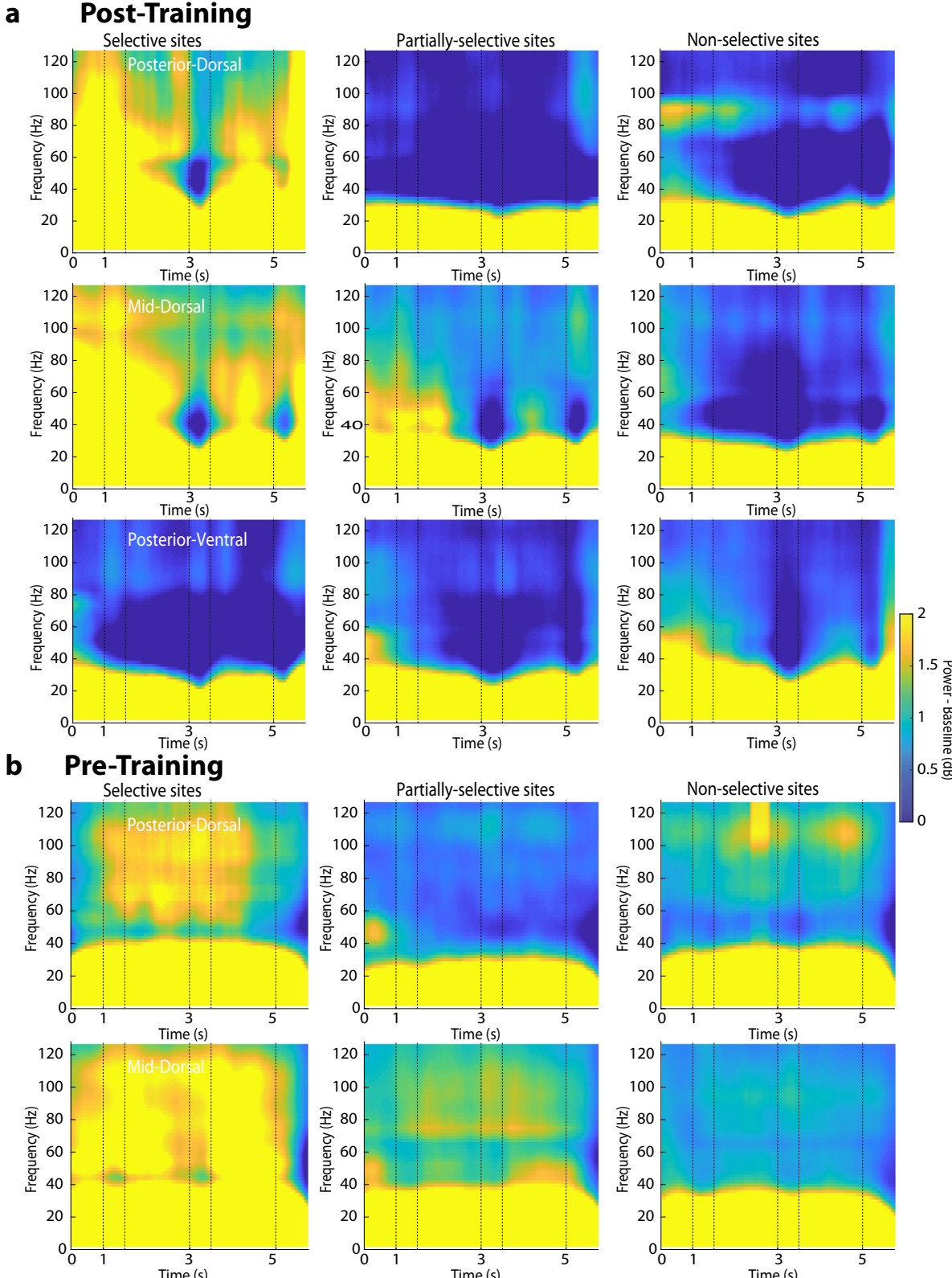

**Fig. 3 Power spectrum of selective vs. non-selective sites in different areas.** Mean LFP induced spectral power shown after (**a**) and prior to training (**b**). Results are shown separately from the posterior-dorsal subregion of the prefrontal cortex ($n = 1079, 3650, 3132$ and $1027, 2234, 2036$ trials for selective, partially-selective sites, non-selective sites, after training and before training, respectively); from the mid-dorsal ($n = 7352, 5716, 8028$ and $3894, 11119, 8490$ trials after and before training); and from the posterior-ventral subregion ($n = 2483, 4670, 13618$ after training). Power is plotted as a function of time, after the subtracting the mean power computed in the inter-trial interval at each frequency. Conventions are the same as in Fig. 2d.

cortex ($F_{2,1475} = 24.7$, $p < 0.001$), with selective sites exhibiting higher gamma power than partially-selective or nonselective sites (post-hoc test, $p < 0.05$). Similarly, a significant main effect of site selectivity was present in the posterior-dorsal prefrontal cortex ($F_{2,475} = 17.4$, $p < 0.001$) with selective sites exhibiting higher gamma power (post-hoc test, $p < 0.05$). In contrast, little gamma power modulation across selective and nonselective sites was present in posterior ventral prefrontal cortex after training. Insufficient data were available prior to training, because very few posterior ventral prefrontal sites exhibited spatial selectivity prior to training[23]. In fact, nonselective sites exhibited higher gamma power compared with partially-selective sites in posterior-ventral cortex (main effect of site selectivity, $F_{2,1250} = 3.5$, $p = 0.03$; post hoc-test comparing nonselective and partial sites, $p < 0.05$). This analysis also confirmed that gamma power decreased after training in individual prefrontal subdivisions. A significant main effect of training phase was present in the mid-dorsal ($F_{1,1475} = 118.2$, $p < 0.001$), posterior dorsal ($F_{1,475} = 21.9$, $p < 0.001$), and posterior ventral prefrontal subdivision ($F_{1,1205} = 29.8$, $p < 0.001$).

The results of the analysis presented so far compared data obtained in animals prior to training, while the monkeys viewed data passively, with data obtained in trained animals while they actively performed the tasks. In that sense, differences between stages may have been due to either the training or the execution of the task. To determine the relative influence of the two factors, we analyzed a dataset of 73 electrode recordings obtained after training for which data were available from both the execution of the task, and from a passive viewing condition, mirroring the trials the monkeys performed prior to being trained. We tested a 3-way model with factors passive or active task; site selectivity; and task epoch. Now, no significant decrease in gamma power was evident for active task performance over passive presentation ($F_{1,705} = 2.63$, $p = 0.11$). Examining the selective sites alone (Supplementary Fig. 5) and performing a repeated measures ANOVA with factors epoch and passive/active task similarly showed no significant main effect of passive vs. active task ($F_{1,150} = 0.24$, $p = 0.63$). The results suggest that training produced enduring effects in prefrontal cortex gamma power and the decrease observed after training was not primarily driven by the active execution of the task.

**Match effect suppresses gamma power.** One of the defining features of the trained task is the judgment on whether the second stimulus has been repeated and constitutes a match or is novel in the trial and constitutes a nonmatch. Many prefrontal neurons respond differently to the same second stimulus depending on whether it appears as a match or nonmatch, in the context of our task and others[24,25]. Generally, responses to a match stimulus are most often suppressed relative to a nonmatch, a phenomenon known as repetition suppression[26], although a strong preference for the nonmatch stimulus was not evident in firing rate, in our dataset (Fig. 2b). We therefore sought to determine how gamma power differed in match and nonmatch trials. We thus averaged power computed in the second delay period for all trials obtained from each electrode and treated as a single observation. Gamma power in nonmatch trials was higher than match trials, when monkeys performed the task, after training (Fig. 4). The effect was only significant for selective sites (paired t-test, $t_{92} = 2.81$, $p = 0.006$); the difference did not reach significance for either partially-selective ($t_{132} = 1.01$, $p = 0.32$) or nonselective sites ($t_{264} = 1.36$, $p = 0.17$). In all cases, the effect was most pronounced in a sub-band of the gamma frequency band, roughly extending from 45 to 85 Hz (indicated with a square in Fig. 4 – though we relied on the a priori selected 45–100 Hz range for our statistical comparisons).

Importantly, the difference in gamma power between match and nonmatch trials was only present after training; no significant difference was present in selective sites prior to training, when monkeys viewed stimuli passively (paired t-test, $t_{45} = -0.30$, $p = 0.76$ in Supplementary Fig. 6). Indeed, the two stimulus presentations prior to training-induced similar increases in gamma band power around the two stimulus presentations (Fig. 2e, around the 1.5 and 3.5 s time points) and no significant main effect of task epoch was present ($F_{4,1883} = 0.13$, $p = 0.97$), unlike the distinct pattern of gamma power observed at each task epoch after training. These results suggest that gamma band power is sensitive to the cognitive operations performed during the task but decrease of gamma power characterizes some important events in the context of the task, such as detection of the match stimulus.

**Shape selectivity is less strongly associated with LFP gamma power.** Analysis so far focused on the spatial stimulus set and spatial working memory task. We recorded neural activity and LFPs using a second stimulus set, differing in shape, before and after training monkeys in the match–nonmatch task using these stimuli (Fig. 1b). Neurons that exhibited selectivity for the shape stimuli were less frequent than spatially selective ones, and therefore fewer selective sites were identified (Fig. 1d). Some nonselective sites in terms of shape information still exhibited selectivity in terms of spatial location. Shape-selective sites were encountered in all prefrontal subdivisions (Supplementary Fig. 7).

As was the case for the spatial stimuli, gamma power was higher in selective over nonselective sites, both before and after training (Fig. 5d, e). We again compared sites, using a 2-way mixed-effects model, with fixed effects including the selectivity status of each site for the shape stimuli (selective, nonselective, or partially-selective) and the task epoch. Selective sites exhibited the highest gamma power than the partially-selective and nonselective sites ($F_{2,1740} = 11.4$, $p < 0.001$; and post-hoc test, $p < 0.05$). Gamma power in the partially-selective and nonselective sites was not significantly different from each other. Strong gamma power in selective sites was evident in all prefrontal subdivisions (Supplementary Fig. 8). Prior to training gamma power again differed between sites depending on whether they were selective or not (2-way model, $F_{2,1104} = 20.1$, $p < 0.001$), with the highest power seen for selective sites. However, in this case, substantial power was seen in nonselective sites. As was also the case for the spatial stimuli, a highly significant decrease of gamma power was evident after training (3-way model, $F_{1,2844} = 97.9$, $p < 0.001$). Gamma power also did not tightly follow the envelope of firing rate of prefrontal neurons (Fig. 5a–c).

The presence of gamma power in sites nonselective for shape raised the possibility that sites responsive to stimuli and displaying spatial selectivity, even if not necessarily selective for shape information ended up determining the site's gamma power. We, therefore, re-grouped nonselective sites for shape, based on their selectivity in the spatial stimulus set, which was tested in a different block of trials (Supplementary Fig. 9). Indeed, gamma power differed in these three groups of sites, ($F_{2,854} = 5.64$ for main effect of site selectivity, $p = 0.003$) and was significantly higher in selective sites compared with the other two groups, and in the partially-selective compared with the nonselective sites (post-hoc test, evaluated at the $\alpha = 0.05$ significance level). When we repeated the same analysis for the pretraining data, we again saw gamma power that differed across sites ($F_{2,599} = 4.0$, $p = 0.018$) with nonselective sites now showing the lowest gamma power. This result too argues against the idea that gamma power is driven by the task being executed; sites responsive to some properties of the stimuli being presented generated robust gamma power, even if that property was not relevant to the task at hand.

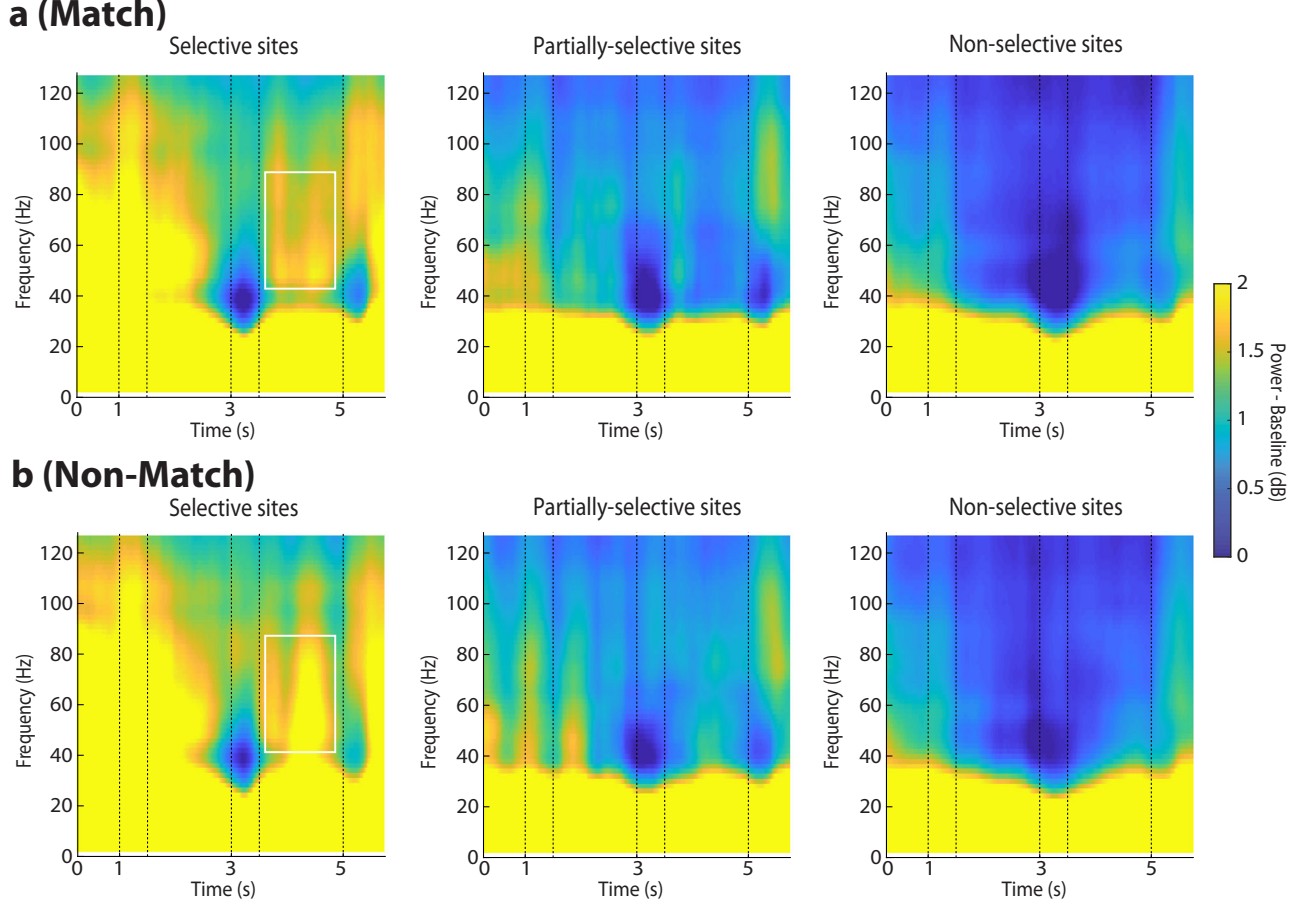

**Fig. 4 Power spectrum of spatial task for match and nonmatch stimuli.** Mean LFP induced spectral power shown separately for match and nonmatch conditions recorded with the spatial stimulus set, after training. **a** Results are shown separately for match responses from selective sites (6821 trials); partially-selective sites (8385 trials) and non-selective sites (17230 trials). **b** Results from non-match trials (n = 6833, 8365 and 17172 trials for selective, partially selective, and non-selective sites, respectively). Conventions are the same as in Fig. 2d.

**Relationship between firing rate and LFP gamma power**. The overall decrease in gamma power after training was puzzling considering that firing rate of selective sites increased after training, both in absolute terms and relative to the prestimulus baseline, and for both the spatial (Fig. 2a) and shape stimulus set (Fig. 5a). To gain insight on the relationship between firing rate and LFP gamma power, we calculated the multiunit firing rate of each site (sum of firing rates of all neurons isolated in this site) at each epoch. We then performed a regression analysis, adding a term of random effects for different animals, to determine how predictive (the logarithm of) firing rate was on gamma power. A positive relationship between firing rate and raw gamma power was evident both before and after training in the spatial task, however, the slope reached significance only after training (Fig. 6b–d). Prior to training while the monkeys viewed the spatial stimuli passively, a positive slope (Fig. 6a) was present for selective sites, (regression analysis, slope=0.95, $t_{226} = 1.15$, $p = 0.25$), partially-selective sites (slope = 0.45, $t_{701} = 0.96$, $p = 0.34$) and nonselective sites (slope = 0.4, $t_{961} = 1.2$, $p = 0.23$). After training, when monkeys performed the spatial task (Fig. 6b), a significantly positive relationship was present in selective sites (slope = 1.73, $t_{466} = 4.67$, $p < 0.001$), partially-selective sites (slope=1.97, $t_{661} = 5.45$, $p < 0.001$) and nonselective sites (slope=0.89, $t_{1361} = 5.21$, $p < 0.001$).

The effects of training we reported were most evident for gamma power after subtracting the baseline – this is in fact what was plotted in all preceding analyses. We therefore tested how

this measure depended on firing rate. Before training (Fig. 6c), when monkeys viewed the spatial stimuli passively, gamma power relative to baseline increased as a function of multiunit firing rate for selective and partially selective sites (regression analysis, slope = 0.41, $t_{216} = 2.2$, $p = 0.03$ for selective sites; slope=0.41, $t_{698} = 4.6$, $p < 0.001$ for partially-selective sites). Interestingly, after training (Fig. 6d), we observed a negative relationship between firing rate and relative gamma power for selective sites (regression analysis, slope = −0.89, $t_{470} = −4.6$, $p < 0.001$). A significant difference in the slope of the gamma power-firing rate relationship was evident for the selective sites between the pretraining and post-training data (t test, $t_{1,682} = 4.97$, $p < 0.001$). The results indicate that although firing rate is predictive of raw gamma power across task conditions and training stages, the relative modulation of gamma power above the baseline is not consistently predicted by firing rate.

**Discussion**
Gamma band oscillations have been implicated in a range of cognitive processes[1,2,27,28]. In the case of working memory, elevated prefrontal gamma band LFP power has been observed in the delay periods of tasks during which information is maintained in memory, relative to baseline[4–6,10]. Furthermore, gamma power has been identified as specific to sites selective for the stimulus information a subject is using for working memory and other tasks, and conversely, sites characterized by gamma power

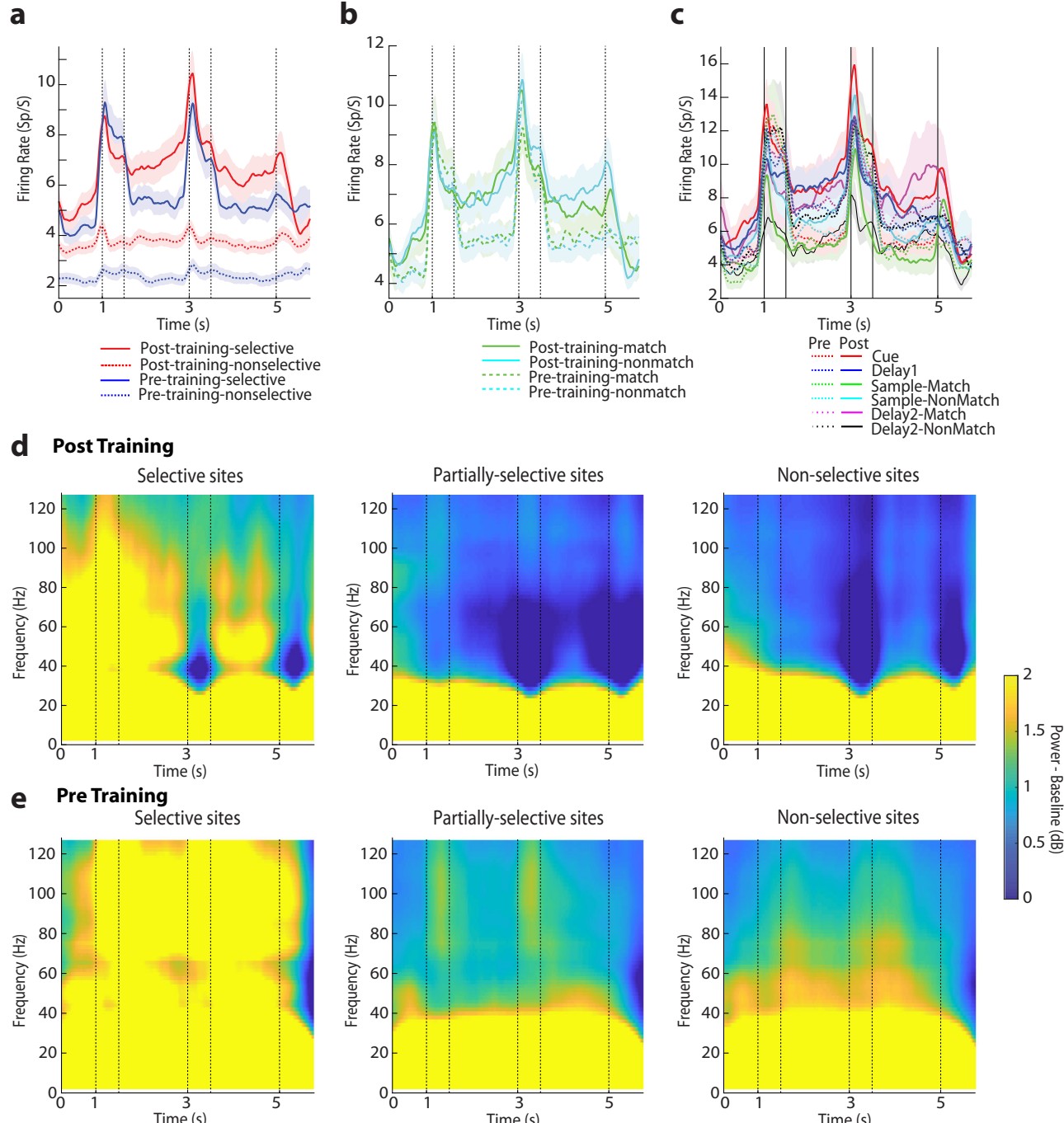

**Fig. 5 Firing rate and spectral power in the shape task. a** Neuronal spiking in the shape stimulus set, prior to and after training. Mean firing rate is shown separately for selective (N = 174 and 241 neurons, prior to training and after the training, respectively); and non-selective sites (N = 465 and 626, respectively). **b** Mean neuronal spiking shown separately for match and nonmatch trials from selective sites (N = 174 and 241 prior to training and after the training respectively). **c** Mean firing rate shown separately for sites that exhibited selectivity during each of the four individual task epoch i.e. first (N = 86 neurons) and (N = 55, 63 neurons) for match and non-match during second stimulus presentation prior to training; N = 91 and (N = 69, 76) neurons after training; and first delay period N = 56 and second delay period N = 36, 35 neurons prior to training; N = 64 and N = 31, 48 after training. **d** Mean LFP induced spectral power recorded with the shape stimulus set from the prefrontal cortex, after training. Spectral power from selective sites (left column), partially-selective sites (middle column) and non-selective sites (right column) is shown separately. N = 4603 trials, 13329 trials, and 20234 trials for selective, partially-selective, and non-selective sites after training, respectively; **e** As in **d**, for recordings prior to training (n = 2784 trials, 10872 trials and, 16442 trials respectively).

contain neurons selective for the stimuli[10,11]. Our study sought to determine to what extent observed patterns of LFP power in trained subjects are shaped by training and performing the task. We indeed confirmed that selective sites exhibit stronger gamma band power during execution of spatial working memory tasks.

Unexpectedly, we found that gamma power relative to baseline was even more so enhanced when animals were naïve about the task rules for decision-making and were only required to view the stimuli passively, across all prefrontal subdivisions tested. Secondly, the relationship between selective sites and gamma

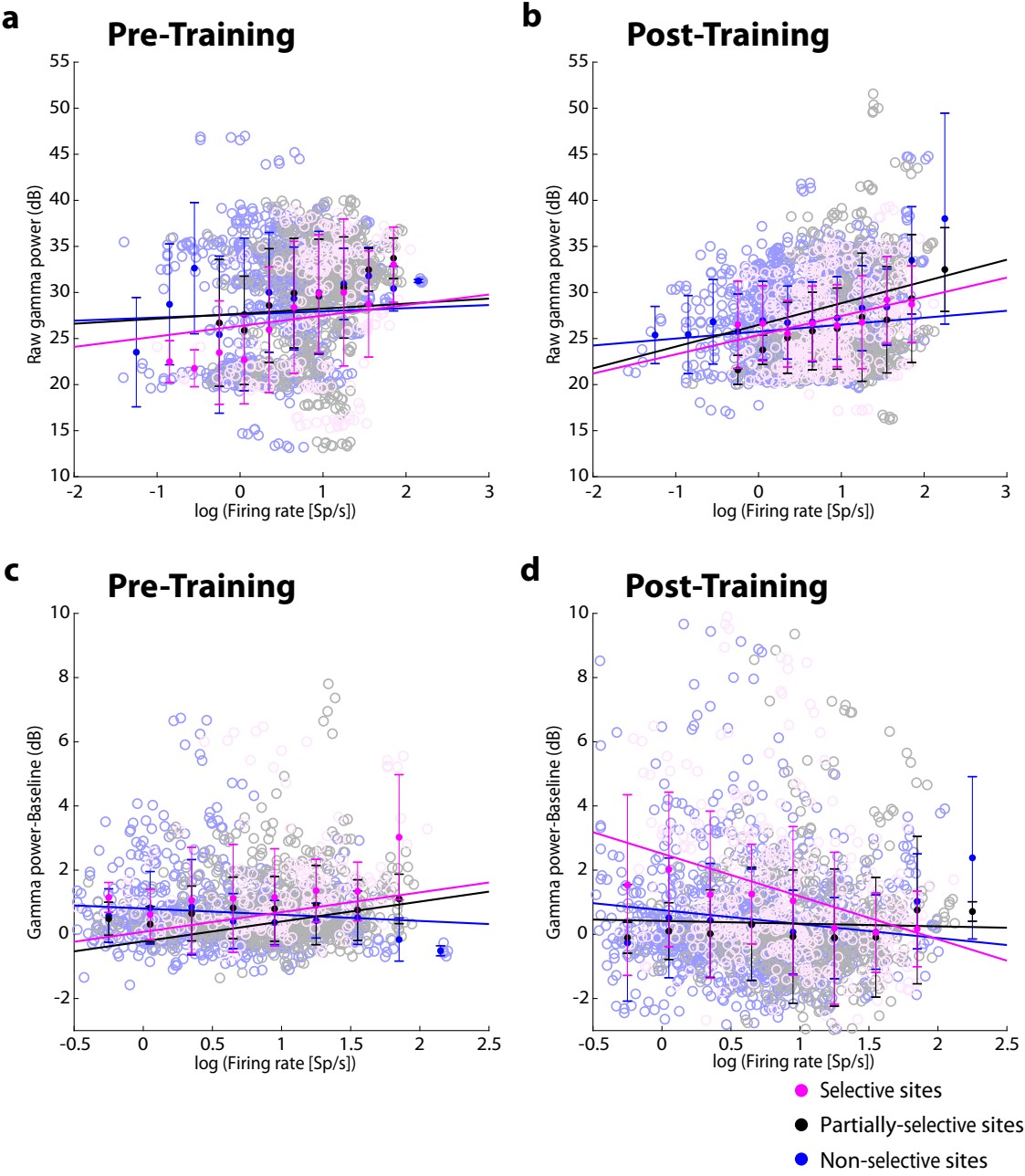

**Fig. 6 Relationship between LFP gamma power and multi-unit firing rate. a** Relationship between raw gamma power and firing rate prior to training is shown for the spatial stimulus set, for selective, partially, and non-selective sites respectively (open circles). Filled circles represent mean power of data averaged in a bin centered on the point, plotted against the mean firing rate of multi-unit records averaged from the same site. Error bars represent standard deviation. Data binned in this fashion are presented for visualization purposes; regression lines were determined based on the raw data, further treating individual monkeys as random effects. **b** Relationship between raw gamma power and firing rate, as computed in *a*, after training. **c** Relationship between gamma power relative to the inter-trial baseline and firing rate, prior to training. **d** Relationship between relative gamma power and firing rate as computed in **c**, after training.

band frequency was less pronounced for shape working memory tasks. This was partly due to the fact that sites selective for the location of a stimulus but not its shape tended to generate higher gamma power, even when spatial location was not necessary in the context of the shape working memory task. Our results suggest that gamma power relative to baseline, represents automatic processing of stimuli and decreases after training as these stimuli are incorporated in the context of cognitive tasks. After training, modulation of gamma power by the execution of cognitive operations in the course of a trial is reliably evident at task critical intervals, e.g., causing it to decrease after the appearance of a

second stimulus that requires a judgment and to become negatively correlated with firing rate.

**Modulation of gamma power during the course of a trial**. A number of changes in gamma power were evident after training to perform the task, including changes during the course of a trial. Whereas prior to training two stimulus presentations elicited nearly identical patterns of LFP spectral power (e.g. in Fig. 2e), after training, a profound modulation of the envelope of power was seen during the presentation of the second stimulus, which

now required a judgment of whether it was the same or not. This finding is consistent with the known engagement of the prefrontal cortex in decision-making processes across a range of tasks and paradigms[29,30]. Tellingly, the presence of this cognitive factor in our task decreased spectral power centered at around 45 Hz (e.g. in Fig. 2d) rather than increase it.

Another related difference in spectral composition centered on gamma power involved trials involving matching and nonmatching presentation of spatial stimuli. Gamma power during the second delay period was greatly reduced in match compared with nonmatch trials. Previous studies have indicated that stimuli presented repeatedly produce decreased responses, a phenomenon referred to as repetition suppression[26]. Repetition suppression has been observed in multiple brain areas and using various methodologies, including single neuron recordings, fMRI, and EEG/MEG[26]. We now show that repetition suppression is evident in LFP gamma power, as well. This effect was tied specifically to the execution of the task; no similar effect was present prior to training. The result was still somewhat unexpected in our data, because in terms of firing rate, approximately equal populations of neurons exhibited preference for a match over a nonmatch spatial stimulus after training, as shown in Fig. 2b, and previously reported[24]. A previous study with a task testing whether a sequence of two stimuli was followed by a matching sequence or not, also did not report decreased gamma power for a match, though their metric was related specifically to "bursting" events in the LFP gamma power[9]. This result suggests that cognitive process during the execution of behavioral tasks do modulate gamma power, however, these do not typically translate into an increase of gamma power.

### Relationship between LFP gamma power and neuronal firing.

Power in the highest frequencies of the LFP spectrum exhibits the strongest correlation with firing rate of units recorded from the same site[31]. It therefore follows that stronger gamma-frequency LFP power is evident in sites where spiking activity is also elevated during working memory[9,32]. In our data, prior to training, a positive relationship with firing rate was present for both raw gamma power and gamma power relative to baseline. Unexpectedly, we found that gamma power decreased after training, during the period of active stimulus maintenance. Strong gamma power tied to stimulus-driven spiking has been widely reported, for example in the primate primary visual[33] and motor cortex[19]. Due to tight correlation between LFP gamma power and spiking activity[18,19] an increase in LFP power might have otherwise been expected after training, when firing rate generally increased particularly in delay periods[34].

The overall decrease in LFP gamma power relative to the baseline we observed after training also appeared to decouple relative power from firing rate. This measure of gamma power was calculated after subtracting the inter-trial power at each frequency. In this sense, it was the stimulus-related gamma power that declined after training and was no longer correlated with firing rate. A potential interpretation of the effect is that a greater percentage of power was tied to stimulus-independent components of firing rate that were evident after training. Indeed, strong evidence exists that training alters neuronal firing in a permanent way[12], with changes evident in baseline firing and stronger nonstimulus-dependent components emerging after training[35].

A number of other factors may account for this finding. Training effects involve changes in trial-to-trial response variability and correlation of firing rates between neurons, both of which typically decline[36,37]. In other words, responses of individual neurons become less variable and more decoupled from nearby neurons, a result generally consistent with the

decline in coordinated activity captured by LFP power we observed after training. Bottom-up processing has been associated with increased gamma power[38,39] and in this sense, finding stronger gamma power in the pretraining LFP is not surprising, considering that bottom-up factors dominated processing prior to learning to perform a cognitive task. It has not been hitherto appreciated that gamma power may appear in the LFP even before training or execution of cognitive tasks.

Our analysis allowed us to characterize the information content of different electrode sites. Sites where neurons selective for spatial stimuli were encountered, exhibited the strongest gamma power in the LFP, in agreement with prior studies[10]. A partially overlapping, though smaller set of these sites also exhibited selectivity for shapes. The same finding also held for these neurons: selective sites were more likely to generate increased gamma power. However, in this case too, the relationship between gamma power and selectivity was not tied to the execution of a task, and robust gamma power was evident for nonshape-selective sites, that exhibited spatial selectivity, even when the latter was irrelevant for the task being executed.

## Methods

Data were analyzed from three male rhesus monkeys (*Macaca mulatta*), ages 5–9 years old, weighing 5–12 kg. None of these animals had any prior experimentation experience at the onset of our study. Monkeys were either single-housed or pair-housed in communal rooms with sensory interactions with other monkeys. All experimental procedures followed guidelines set by the U.S. Public Health Service Policy on Humane Care and Use of Laboratory Animals and the National Research Council's Guide for the Care and Use of Laboratory Animals and were reviewed and approved by the Wake Forest University Institutional Animal Care and Use Committee under protocol number A14-196.

Monkeys sat with their heads fixed in a primate chair while viewing a monitor positioned 68 cm away from their eyes with dim ambient illumination and were required to fixate in a 0.2° white square appearing in the center of the screen. During each trial, the animals were required to maintain fixation on a 0.2° white square appearing in the center of the screen while visual stimuli were presented either at a peripheral location or over the fovea, in order to receive a liquid reward (typically fruit juice). Any fixation break immediately terminated the trial and no reward was given. Eye position was monitored throughout the trial using a non-invasive, infrared eye position scanning system (model RK-716; ISCAN, Burlington, MA). The system achieved a < 0.3° resolution around the center of vision. Eye position was sampled at 240 Hz, digitized and recorded. The visual stimulus display, monitoring of eye position, and synchronization of stimuli with neurophysiological data was performed with in-house software implemented on the MATLAB environment (Mathworks, Natick, MA), utilizing the Psychophysics Toolbox[40].

### Behavioral task

*Pretraining task.* Following a brief period of fixation training and acclimation to the stimuli, monkeys were required to fixate on a center position while stimuli were displayed on the screen. The stimuli shown in the pretraining, passive, spatial task consisted of white 2° squares, presented in one of nine possible locations arranged in a 3 × 3 grid with 10° of distance between adjacent stimuli. The stimuli shown in the pretraining passive shape task consisted of white 2° geometric shapes drawn from a set comprising a circle, diamond, the letter H, the hashtag symbol, the plus sign, a square, a triangle, and an inverted Y-letter. The stimuli analyzed here were always presented at the center location of the 3 × 3 grid.

Presentation began with a fixation interval of 1 s where only the fixation point was displayed, followed by 500 ms of stimulus presentation (referred to hereafter as cue), followed by a 1.5 s "delay" interval where, again, only the fixation point was displayed. A second stimulus (referred to hereafter as sample) was subsequently shown for 500 ms. In the spatial task, this second stimulus would be either identical in location to the initial stimulus, or diametrically opposite the first stimulus. In the shape task, this second stimulus would appear in the same location to the initial stimulus and would either be an identical shape or the corresponding nonmatch shape (each shape was paired with one nonmatch shape). Only one nonmatch stimulus was paired with each cue, so that the number of match and nonmatch trials were balanced in each set. In both the spatial and shape task, this second stimulus display was followed by another "delay" period of 1.5 s where only the fixation point was displayed. The location and identity of stimuli was of no behavioral relevance to the monkeys during the "pretraining" phase, as fixation was the only necessary action for obtaining reward.

*Training.* Once recordings were obtained in the pretraining phase, which lasted 1-6 months in the three monkeys (Supplementary Table 2), the monkeys were then

trained to perform working memory tasks that involved the presentation of identical stimuli as the spatial and shape tasks during the "pretraining" phase. Training lasted for 8.5, 8, and 16 months in the three monkeys (ADR, ELV, and NIN, respectively). The animals were allowed to reach asymptotic performance before recordings began. The three subjects achieved different levels of performance. Mean (and S.E.M.) proportion of correct trials in the task among sessions analyzed here was $0.62 \pm 0.06$, $0.95 \pm 0.05$, and $0.83 \pm 0.06$, for the three subjects, respectively.

*Post-training task*. Now monkeys were required to remember the spatial location and/or shape of the first presented stimulus, and report whether the second stimulus was identical to the first or not, via saccading to one of two target stimuli (green for matching stimuli, blue for nonmatching). Each target stimulus could appear at one of two locations orthogonal to the cue/sample stimuli, pseudo-randomized in each trial.

**Surgery and neurophysiology**. A 20 mm diameter craniotomy was performed over the PFC and a recording cylinder was implanted over the prefrontal cortex. The location of the cylinder was visualized through anatomical magnetic resonance imaging (MRI) and stereotaxic coordinates post-surgery. Electrode penetrations were mapped onto the cortical surface. We identified 6 lateral PFC regions: a posterior-dorsal region that included area 8 A, a mid-dorsal region that included area 8B and area 9/ 46, an anterior-dorsal region that included area 9 and area 46, a posterior-ventral region that included area 45, an anterior-ventral region that included area 47/12, and a frontopolar region that included area 10[41]. Only posterior dorsal, mid-dorsal and posterior-ventral areas were sufficiently sampled for between subdivision-comparisons.

**Neural recordings**. Recordings were performed from the prefrontal cortex both before and after training in the working memory tasks. Subsets of the data presented here were previously used to determine the individual properties of neurons in the posterior-dorsal, mid-dorsal, anterior-dorsal, posterior-ventral, and anterior-ventral PFC subdivisions[17]. Extracellular recordings were performed with multiple microelectrodes that were either glass- or epoxylite-coated tungsten, with a 250 μm diameter and 1–4 MΩ impedance at 1 kHz (Alpha-Omega Engineering, Nazareth, Israel). A Microdrive system (EPS drive, Alpha-Omega Engineering) advanced arrays of up to 8-microelectrodes, spaced 0.2–1.5 mm apart, through the dura and into the PFC. The signal from each electrode was amplified and band-pass filtered between 500 Hz and 8 kHz while being recorded with a modular data acquisition system (APM system, FHC, Bowdoin, ME). Waveforms that exceeded a user-defined threshold were sampled at 25 μs resolution, digitized, and stored for off-line analysis. Neurons were sampled in an unbiased fashion, collecting data from all units isolated from our electrodes, with no regard to the response properties of the isolated neurons. We used the KlustaKwik algorithm[42] to sort recorded spike waveforms into separate units. The program applies principal component analysis on waveforms to create clusters of units with similar waveforms; all clusters were inspected by the investigators before being accepted as units. We reconstructed multiunit records by adding together all sorted units in each electrode.

**LFP recordings**. LFP data were acquired in the aforementioned areas of the PFC, both before and after training in each working memory task, through extracellular recordings with the same microelectrodes used to record single-neuron activity. The signal from each electrode was amplified and band-pass filtered between 0.5 and 200 Hz for LFP processing, with the same modular data acquisition system (APM system, FHC, Bowdoin, ME).

**Data analysis**

*Neural data*. Data analysis was implemented with the MATLAB computational environment (R2014-2021, Mathworks, Natick, MA). A trial-averaged peristimulus time histograms (PSTHs) for illustrations were calculated by convolving the spiking events a 50 ms steps apart. Neurons were identified to be selective in any task epoch by virtue of significantly different responses to the spatial location (or shape) of the stimulus, using a one-way ANOVA test performed on the firing rates of each neuron obtained across trials during the task epoch in question. To avoid false positives among neurons with a handful of spikes, we also required that a selective neuron exhibit a firing rate of at least 2 spikes/s for its best stimulus location or shape, in the task period that the ANOVA test indicated a significant main effect.

*LFP analysis*. LFP recording were preprocessed by using MATLAB code in MATLAB R2019a (MathWorks) and the FieldTrip toolbox[43]. A bandpass filter between 0.5 and 200 Hz with a zero-phase sixth-order Butterworth filter were used on single-trial LFP traces; we also used a notch filter at 60 Hz with a bandwidth of 0.1 Hz to remove the power line artifacts. Further, single-trial LFP traces underwent artifact rejection. The Chronux package[44] was used for time-frequency analysis. We used a multitaper method to perform a power spectrum analysis of LFPs. The spectrogram of each single trial between 0.5 and 128 Hz was computed with 8 tapers in 500 ms time windows; the spectrograms were estimated with a

temporal resolution of 2 ms. We also used the mean filter corresponding to 1.95 Hz and 20 ms for smoothing the spectrogram of each single trial. We relied on induced power of the LFP in all of our analysis, which is computed by first performing a power computation in each trial and then power across trials is averaged. Induced power thus determines power at specific frequencies that may not necessarily be synchronized with specific task events across trials. Power was expressed relative to the mean power recorded during the inter-trial interval. Time-resolved plots (spectrograms) were constructed and plotted after dividing the power of the signal by the mean inter-trial interval power at each frequency (which is equivalent to subtracting the baseline power in logarithmic, dB, scale). Raw LFP signal was represented as dB levels of the data acquisition system's analog-to-digital unit.

*Statistical analysis*. Statistical testing of differences between conditions was performed in the following fashion. First we calculated power across an entire epoch: fixation period, cue presentation, first delay, sample presentation, second delay (rather than at every time point, as illustrated in spectrograms). Secondly, we averaged power values in these epochs from all trials of every electrode site, essentially treating each LFP site as one observation. We then constructed a 2-way or 3-way mixed-effects linear model with fixed-effects terms representing the task epoch, site selectivity, and training status, and random effects term for each monkey as follows:

gammapower ~ epoch * selectivity + (1|Monkey)
gammapower ~ epoch * selectivity * training + (1|Monkey)

Similarly, the regression analyses examined the following mixed-effects linear model with firing rate as a continuous fixed-effects variable and random intercepts for each monkey.

gammapower ~ firingrate + (1|Monkey)

In every case, the analysis was performed for the gamma frequency band, defined as 45-100 Hz; effects of training on other LFP properties have been reported elsewhere[12,13].

These models were implemented in the R computational environment (v4.2.2) using the lmer R package and Rstudio. F-statistics and p-values were calculated with Satterthwaite's or Kenward-Roger's method for degrees-of-freedom determination. F-statistics were calculated based on the glme model (lmerTest R package). Post-hoc pairwise comparisons on any significant main effects were performed with Tukey's method using the emmeans function (emmeans R package).

**Reporting summary**. Further information on research design is available in the Nature Portfolio Reporting Summary linked to this article.

## Data availability

This paper does not report original code. Data used for the analysis and figures will be deposited at Mendeley (https://doi.org/10.17632/2ggs6yzyz2.1). Any additional information required to reanalyze the data reported in this paper is available from lead contact upon request.

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

## Acknowledgements

We wish to thank Wenhao Dang for valuable comments on the manuscript. Research reported in this paper was supported by the National Eye Institute of the National Institutes of Health under award number R01 EY017077.

## Author contributions

Conceptualization, C.C.; Investigation C.C., B.S.; Data Curation, C.C., B.S.; Software B.S.; Formal Analysis B.S., C.C., and Z.W.; Visualization, B.S.; Writing-Original Draft, C.C. and B.S.; Writing-Review & Editing, C.C., B.S. Z.W.; Supervision, C.C.; Project Administration, C.C.; Funding Acquisition C.C.

## Competing interests

The authors declare no competing interests.
