## [Peer Review File · Communications Biology]

Reviewers' comments:

Reviewer #1 (Remarks to the Author):

Brief Summary:

In this manuscript, Singh et al. investigate the effect of training and task context on LFP power in the gamma frequency range in prefrontal cortex. They trained monkeys on two tasks, a spatial matching task and a shape matching task, and looked at LFP power (45-100Hz) during passive viewing before training and during the same time within an active response version of the task, after training, across recording sites. Recording sites were categorized as selective, partially selective, or non-selective by neuronal firing rate responses to the task stimuli. They found gamma power increased relative to baseline during the stimulus presentation period, when compared to baseline. They also found that gamma power was the highest for recording sites labeled selective (i.e. all neurons were selective for the task stimuli) both before and after training. However, they also found the effect of stimulus presentation on gamma power was reduced after task training, even for selective sites. Lastly, they found that gamma power was different between match versus nonmatch stimuli after training on the active response task. The authors assert that gamma power is not driven by the task being executed, rather, stimulus processing.

Overall Impression:

Overall, this manuscript provides a useful set of observations about gamma power in prefrontal cortex but needs some major revisions of the methods and conclusions drawn from the data. The conceptual novelty is a bit thin and there appears to be a major confound regarding the comparison before and after training on the active response task. Specific comments are listed below.

Specific comments:

1. Major concern: "pre-training" vs. "post-training". I have major concerns about the use of the names "pre-training" and "post training" and the comparison being made, which is not about learning, but passive versus active use of stimuli for decision-making. Critically, I found "pre-training" and "post-training" to be extremely misleading, as the descriptions in the Abstract, Introduction, Results and Discussion section of the text made it sound like it was before and after the monkeys had learned the spatial and shape tasks, (e.g. Lines 252-253: "...we found that gamma power relative to baseline was even more so enhanced in naïve animals, prior to being trained to perform the task at all, across all prefrontal subdivisions tested..." but according to the Methods, "pre-training" was actually "passive viewing." (and Fig. 1 legend). Animals were overtrained in both tasks, as the same stimuli are used every day and the tasks do not change. From a visual familiarity perspective, the monkeys are not naïve to the passive viewing task, rather, they are overtrained on that task. There is a huge emphasis on learning here and that seems misleading in many ways. The discussion does mention "Our study sought to determine to what extent patterns ...are shaped by training and performing the task" (Lines 248-249). This seems like a more appropriate framing for the methods used.

1a.) Minor related concern: It seems an open chamber and multiple single electrodes were used to collect data but the methods are unclear. First, the time scales for trainings and recordings are unclear but critical to the understanding of what "same sites" means for comparison of LFP power pre- and post- training. Could you please add to the methods how you verified "same sites" for an unbiased comparison?

1b.) How long was "pre-training" relative to "post", #s of trials, sessions?

1c.) It is not clear to me when animals were naïve and when they were trained. They were first trained on the passive viewing task, then naïve to the active task, and then at some point "trained" on the active task (i.e. stable behavior). If there was some transition from naïve to "trained," it seems there should be a performance criterion or some metric related to task performance to demonstrate

the difference between these time periods and to give a sense of scale for this time difference (days? Weeks? Months?). And how does this compare across monkeys?

1d.) The data from 3 monkeys is presented as a single population. Did the animals perform identically well on the active task? It would be useful to go into the behavior a bit more (or reference it more clearly if there are figures in another paper) to clarify the validity of pooling across animals in this way. It would still be useful to see a breakdown of how much data came from each monkey. i.e. was just one monkey driving the majority of the effects, or had the majority the selective units and thus signals used in final analyses?

1e.) It would be useful to see a breakdown of the data somewhere ie. number of days, sessions, criterion for learning performance, number of trials, contribution of each monkey to the pooled data set, etc. How many sessions of data were collected, used, omitted, how many from each monkey, etc? How many electrodes per session? There appear to be no details on the details of the dataset except the selectivity breakdowns and area breakdowns. They could go in the Methods.

2. Major concern: There is a confound of comparing data from passive and active tasks and calling it a learning effect. It is not clear that the results would be the same if the comparison had been passive pre- and passive post- . To make a stronger claim that is not confounded by the performance of the task, the authors could've designed the experiment to compare passive pre-training and passive post-training. While I understand that the first part of the trial is the same in both tasks, there is a confound of comparing a passive task to a decision-making task with the same stimuli in this experiment that somewhat undermines the novelty of the result. There are countless papers showing decision-related information in prefrontal cortex and changes related to training, so it is unclear to me what this comparison is really adding to what we already know about the role of prefrontal cortex in decision-making, as the authors are showing changes in gamma between match and mismatched stimuli during a decision-making task when the animal is forming a choice. The discussion currently reads as sort of a list of observations of behavior with gamma, which makes the overall argument seem weaker than it otherwise could be. Perhaps the authors could address how their findings enhance our understanding of the role of prefrontal cortex in passive processing of stimuli versus active use of them.

3. Major concern: conceptual novelty. It remains unclear why the main message about selectivity and gamma is novel, as there are many papers that have shown gamma is thought to be related to spiking activity (e.g. Berens et al., 2008; Waldert et al., 2013; Ahmadi et al., 2021)

Reviewer #2 (Remarks to the Author):

The authors investigated the gamma-band power modulation in cortical LFP during working memory tasks in monkeys. Although their scientific question is sound and recorded data are very precious, their analytical methods need to be more valid. Specifically, I have a serious concern regarding the statistical approach that the authors are taking.

It is necessary to take a look at the distributions of the data. Justification for the use of ANOVA and Tukey tests is needed. Are the data all Gaussian? If not, the authors should go for non-parametric statistical tests. This is an elementary standard in terms of data analysis, and the lack of information is critical – it is also put in Communications Biology submission guidelines. At the moment, it is hard to provide more in-depth review comments on the results and discussion.

How did the authors aggregate the data from three monkeys? If the authors aggregated and concatenated them across monkeys, it would cause a bias toward specific animals. Also, please provide the information regarding how data were obtained from three monkeys (e.g., how many neurons and sites). If the authors are not familiar with the bias issue, please see Yu et al. Beyond t test and ANOVA: applications of mixed-effects models for more rigorous statistical

analysis in neuroscience research. Neuron Volume 110, Issue 1, 5 January 2022, Pages 21-35
<https://doi.org/10.1016/j.neuron.2021.10.030>

It might not be necessary to report exact p values small than 0.001. It is not informative to show p values such as $p=5.7E-21$ and $p=1.36E-14$ with a inconsistent significant digits.

"p" should be italicized.

Please give a reference or more details for the statement below (line 416-418)

> A semi-automated cluster analysis relied on the KlustaKwik algorithm, which applied principal component analysis of the waveforms to sort recorded spike > waveforms into separate units.

COMMSBIO-22-4280-T
Singh et al.
Response to reviewers

Reviewer #1:

0. Overall, this manuscript provides a useful set of observations about gamma power in prefrontal cortex but needs some major revisions of the methods and conclusions drawn from the data. The conceptual novelty is a bit thin and there appears to be a major confound regarding the comparison before and after training on the active response task. Specific comments are listed below.

Response: We appreciate the reviewer's evaluation. We have revised extensively to address all concerns raised.

1. Major concern: "pre-training" vs. "post-training". I have major concerns about the use of the names "pre-training" and "post training" and the comparison being made, which is not about learning, but passive versus active use of stimuli for decision-making. Critically, I found "pre-training" and "post-training" to be extremely misleading, as the descriptions in the Abstract, Introduction, Results and Discussion section of the text made it sound like it was before and after the monkeys had learned the spatial and shape tasks, (e.g. Lines 252-253: "...we found that gamma power relative to baseline was even more so enhanced in naïve animals, prior to being trained to perform the task at all, across all prefrontal subdivisions tested..." but according to the Methods, "pre-training" was actually "passive viewing." (and Fig. 1 legend). Animals were overtrained in both tasks, as the same stimuli are used every day and the tasks do not change. From a visual familiarity perspective, the monkeys are not naïve to the passive viewing task, rather, they are overtrained on that task. There is a huge emphasis on learning here and that seems misleading in many ways. The discussion does mention "Our study sought to determine to what extent patterns ...are shaped by training and performing the task" (Lines 248-249). This seems like a more appropriate framing for the methods used.

Response: The reviewer's point is well taken. We now explicitly point out throughout the text that the two stages differed in that monkeys were trained and executed the task after training, whereas they viewed stimuli passively prior to training. Additionally, we analyzed results obtained after training from sessions in which the monkeys viewed the stimuli passively, as they did prior to training. Differences in gamma power were generally minor for these two conditions, suggesting that training to perform the task produced enduring changes in LFP structure.

2. Minor related concern: It seems an open chamber and multiple single electrodes were used to collect data but the methods are unclear. First, the time scales for trainings and recordings are unclear but critical to the understanding of what "same sites" means for comparison of LFP power pre- and post- training. Could you please add to the methods how you verified "same sites" for an unbiased comparison?

Response: The reviewer is correct; we used recordings with movable electrodes through implanted chambers to record before and after training. We did not wish to suggest that electrodes sampled the exact same penetrations, several months apart. We unfortunately used the term "sites" to also refer to the prefrontal subdivisions sampled. ("A 20 mm diameter craniotomy was performed over the PFC and a recording cylinder was implanted over the site", in the original article). We now use "site" only to refer to electrode penetration and have revised the text to indicate that the same prefrontal subdivisions were sampled before and after training.

3. How long was “pre-training” relative to “post”, #s of trials, sessions?

Response: We now provide this information in the Methods and Supplementary Table 2.

4. It is not clear to me when animals were naïve and when they were trained. They were first trained on the passive viewing task, then naïve to the active task, and then at some point “trained” on the active task (i.e. stable behavior). If there was some transition from naïve to “trained,” it seems there should be a performance criterion or some metric related to task performance to demonstrate the difference between these time periods and to give a sense of scale for this time difference (days? Weeks? Months?). And how does this compare across monkeys?

Response: We have added a section in the Methods to better describe the training phase. Training lasted for 8.5, 8, and 16 months in the three monkeys (ADR, ELV, and NIN, respectively). The animals were allowed to reach asymptotic performance before recordings began. Performance information is provided.

5. The data from 3 monkeys is presented as a single population. Did the animals perform identically well on the active task? It would be useful to go into the behavior a bit more (or reference it more clearly if there are figures in another paper) to clarify the validity of pooling across animals in this way. It would still be useful to see a breakdown of how much data came from each monkey. i.e. was just one monkey driving the majority of the effects, or had the majority the selective units and thus signals used in final analyses?

Response: In view of the comments of also reviewer #2 on this issue, we now present data separately from each monkey in Supplementary Figure S3. We have also refined our analysis and we now present results of a mixed-effects model, in which individual monkeys are treated as random effects.

6. It would be useful to see a breakdown of the data somewhere ie. number of days, sessions, criterion for learning performance, number of trials, contribution of each monkey to the pooled data set, etc. How many sessions of data were collected, used, omitted, how many from each monkey, etc? How many electrodes per session? There appear to be no details on the details of the dataset except the selectivity breakdowns and area breakdowns. They could go in the Methods.

Response: We now present these data in tables S1 and S2.

7. Major concern: There is a confound of comparing data from passive and active tasks and calling it a learning effect. It is not clear that the results would be the same if the comparison had been passive pre- and passive post-. To make a stronger claim that is not confounded by the performance of the task, the authors could've designed the experiment to compare passive pre-training and passive post-training. While I understand that the first part of the trial is the same in both tasks, there is a confound of comparing a passive task to a decision-making task with the same stimuli in this experiment that somewhat undermines the novelty of the result. There are countless papers showing decision-related information in prefrontal cortex and changes related to training, so it is unclear to me what this comparison is really adding to what we already know about the role of prefrontal cortex in decision-making, as the authors are showing changes in gamma between match and mismatched stimuli during a decision-making task when the animal is forming a choice. The discussion currently reads as sort of a list of observations of behavior with gamma, which makes the overall argument seem weaker than it otherwise could be. Perhaps the

authors could address how their findings enhance our understanding of the role of prefrontal cortex in passive processing of stimuli versus active use of them.

Response: As noted above, we now present data from such a control, post-training passive experiment. We do acknowledge the well-known role of the prefrontal cortex in decision making and agree entirely with the reviewer's point that our findings enhance our understanding of the role of prefrontal cortex in passive processing of stimuli versus active use of them. We now cast our main finding as the existence of strong gamma power prior to training, a condition which is unambiguous in terms of task (see also response to the next comment).

8. Major concern: conceptual novelty. It remains unclear why the main message about selectivity and gamma is novel, as there are many papers that have shown gamma is thought to be related to spiking activity (e.g. Berens et al., 2008; Waldert et al., 2013; Ahmadi et al., 2021)

Response: The reviewer's comment made us realize that we failed to convey our main message effectively. Our intended point is that gamma power is present in the prefrontal cortex regardless of task execution, even in naïve monkeys viewing stimuli passively. In that sense, gamma power increase is not a signature of executing a working memory, or other cognitive task. We believe this is a novel finding in direct conflict with the dominant viewpoint in the field about the role of gamma, expressed for example in Miller et al. *Neuron*, 2018. We wished to report that our dataset replicated the effect of elevated gamma power in selective sites, to show that our dataset was not unique or aberrant in that way. We have now rephrased our introduction and discussion to better make this point. We also cite the papers listed above and provide a broader context. As pointed out by the latter two authors, gamma power of the LFP is tightly coupled with spiking, and it is additionally unexpected that gamma power would decline after training, when firing rate generally increased.

Reviewer #2 (Remarks to the Author):

0. The authors investigated the gamma-band power modulation in cortical LFP during working memory tasks in monkeys. Although their scientific question is sound and recorded data are very precious, their analytical methods need to be more valid. Specifically, I have a serious concern regarding the statistical approach that the authors are taking.

Response: We appreciate the reviewer's evaluation. We have repeated our analysis to address statistical issues more rigorously.

1. It is necessary to take a look at the distributions of the data. Justification for the use of ANOVA and Tukey tests is needed. Are the data all Gaussian? If not, the authors should go for non-parametric statistical tests. This is an elementary standard in terms of data analysis, and the lack of information is critical – it is also put in Communications Biology submission guidelines. At the moment, it is hard to provide more in-depth review comments on the results and discussion.

Response: The reviewer's point is well taken. We now show the full distributions of power data in supplementary figure S2. Because power data are essentially log transformed (baseline subtracted from raw power and expressed in dB units) gaussian distributions fitted the data well.

2. How did the authors aggregate the data from three monkeys? If the authors aggregated and concatenated them across monkeys, it would cause a bias toward specific animals. Also, please provide the information regarding how data were obtained from three monkeys (e.g., how many neurons and

sites). *If the authors are not familiar with the bias issue, please see Yu et al. Beyond t test and ANOVA: applications of mixed-effects models for more rigorous statistical analysis in neuroscience research. Neuron Volume 110, Issue 1, 5 January 2022, Pages 21-35 <https://doi.org/10.1016/j.neuron.2021.10.030>*

Response: We are grateful to the reviewer for this suggestion which allowed us to improve the rigor of our analysis considerably. We now provide more details about the sample size and its distribution across animals and areas in Supplementary Tables 1 and 2. We have adopted mixed-effects models and analyze the results accordingly. Figure 6 is also revised to reflect changes to the models.

2d. *It might not be necessary to report exact p values small than 0.001. It is not informative to show p values such as $p=5.7E-21$ and $p=1.36E-14$ with a inconsistent significant digits. "p" should be italicized.*

Response: We have revised accordingly.

2e. *Please give a reference or more details for the statement below (line 416-418) > A semi-automated cluster analysis relied on the KlustaKwik algorithm, which applied principal component analysis of the waveforms to sort recorded spike > waveforms into separate units.*

Response: We now provide reference and explain in more detail how this method was applied and how multi-unit records were reconstituted.

REVIEWERS' COMMENTS:

Reviewer #1 (Remarks to the Author):

The authors have addressed my comments and concerns adequately in their revision.

Reviewer #2 (Remarks to the Author):

I believe that the authors have effectively addressed all my concerns, resulting in a significant improvement of the paper.

COMMSBIO-22-4280A
Singh et al.
Response to reviewers

Reviewer #1:

0. The authors have addressed my comments concerns adequately in their revision.

Response: We appreciate the reviewer's evaluation.

Reviewer #2 (Remarks to the Author):

0. I believe that the authors have effectively addressed all my concerns, resulting in a significant improvement of the paper.

Response: We appreciate the reviewer's evaluation.